# Comparison of Venous and Capillary Sampling in Oral Glucose Testing for the Diagnosis of Gestational Diabetes Mellitus: A Diagnostic Accuracy Cross-Sectional Study Using Accu-Chek Inform II

**DOI:** 10.3390/diagnostics10121011

**Published:** 2020-11-26

**Authors:** Sofia Nevander, Eva Landberg, Marie Blomberg, Bertil Ekman, Caroline Lilliecreutz

**Affiliations:** 1Department of Obstetrics and Gynecology, Linköping University, SE-581 85 Linköping, Sweden; marie.blomberg@regionostergotland.se (M.B.); Caroline.Lilliecreutz@regionostergotland.se (C.L.); 2Department of Biomedical and Clinical Sciences, Linköping University, SE-581 85 Linköping, Sweden; eva.landberg@regionostergotland.se; 3Department of Clinical Chemistry, Linköping University, SE-581 85 Linköping, Sweden; 4Department of Endocrinology, Linköping University, SE-581 85 Linköping, Sweden; Bertil.Ekman@regionostergotland.se; 5Department of Health, Medicine and Caring Sciences, Linköping University, SE-581 85 Linköping, Sweden

**Keywords:** antenatal care, gestational diabetes mellitus, pregnancy, OGTT, fasting glucose, capillary sampling

## Abstract

Gestational diabetes mellitus (GDM) is a common complication with negative impacts on mother and child. The primary aim of this study was to examine whether plasma glucose cutoffs for GDM diagnosis based on venous sampling can be replaced by cutoffs based on capillary sampling. A prospective cross-sectional study was performed at an antenatal care clinic including 175 pregnant women undergoing an oral glucose tolerance test (OGTT). Duplicate samples were collected by capillary and venous puncture while fasting and 1 h and 2 h after an OGTT. Both samples were analyzed on Accu-Chek Inform II. The cutoffs for a GDM diagnosis using capillary samples were corrected from 5.1 to 5.3 mmol/L for the fasting sample, from 10.0 to 11.1 mmol/L for the 1 h sample, and from 8.5 to 9.4 mmol/L for the 2-h sample using half of the dataset. Applying these cutoffs to the remaining dataset resulted in a sensitivity, specificity, and accuracy of 85.0%, 95.0%, and 90.3%, respectively, with a positive predictive value (PPV) of 83%, an negative predictive value (NPV) of 96%, and a positive negative likelihood ratio (LHR) of 16.4 using capillary sampling for the GDM diagnosis at fasting and 2-h after. Corrected cutoffs and capillary samples can be used for the diagnosis of GDM with maintained diagnostic accuracy using Accu-Chek Inform II.

## 1. Introduction

The Hyperglycemia and Adverse Pregnancy Outcomes (HAPO) study from 2008 found a strong continuous association between maternal glucose levels and the incidence of adverse obstetric and perinatal outcomes, e.g., increased risk for cesarean section and newborns being large for gestational age [1,2,3]. Gestational Diabetes Mellitus (GDM) is one of the most common complications during pregnancy, and the prevalence is rising worldwide, with 1% to 28% of pregnancies being affected [4]. The International Diabetes Federation (IDF) estimates that one in six live births (16.8%) are to women with some form of hyperglycemia in pregnancy, and the majority (84%) are related to GDM [2,5]. In the last consensus statement by the International Association of the Diabetes and Pregnancy Study Groups (IADPSG) [2,5], GDM is diagnosed if one or more of the following glucose values is met or exceeded: a fasting venous P-glucose ≥5.1 mmol/L and/or a 1 h value ≥10.0 mmol/L and/or 2 h value ≥8.5 mmol/L post-75 g oral glucose tolerance test (OGTT).

The cutoffs from the HAPO study were based on venous P-glucose as well as the recommendations from IADPSG. However, they do not specify which method to be used to analyze P-glucose [2,5,6]. Some studies indicate that the glucose levels in capillary and venous plasma are not interchangeable, especially not in the non-fasting condition [7,8], but there is controversy about whether capillary plasma samples can be used for diagnosing GDM [9]. There are several advantages with capillary sampling, as it is easier to perform and requires simpler equipment.

Point-of-care testing (POCT) is an attractive option compared to sending the test to a laboratory. POCT has advantages considering cost and convenience and offers the immediate availability of results and the opportunity to measure P-glucose from both venous and capillary samples [10]. Not all POCT methods have enough analytical accuracy to be used in the diagnosis of diabetes. However, Accu-Chek Inform II has been proven to fulfil adequate performance for diagnostic use [11,12]. To date, there have been no studies comparing P-glucose values in capillary and venous samples from pregnant women before and during OGTT, where the same analytical instrument has been used for both sample types.

We hypothesized that capillary sampling using Accu-Chek Inform II can be applied for the diagnosis of GDM.

The primary aim was to examine whether plasma glucose cutoffs for GDM diagnosis based on venous sampling can be replaced by cutoffs based on capillary sampling in an antenatal setting without compromising diagnostic accuracy.

Secondary aims were to evaluate the necessity to continue OGTT after taking fasting samples and to investigate the additional number of women who receive the diagnosis of GDM at 1 h and 2 h.

## 2. Materials and Methods

Pregnant women attending the antenatal care clinic between March 2017 and January 2019 in Linköping, Sweden and who had been recommended to take an OGTT were asked to participate in this study when the total workload allowed it.

The criteria for undergoing an OGTT in gestational weeks (GW) 28–29 were body mass index (BMI) ≥30, previous neonate with a birth weight of ≥4500 g, family history of type 2 diabetes, previously intrauterine fetal death, polyhydramnion, accelerated fetal growth according to the symphysis-fundal height, and non-European origin. If a woman had GDM in a previous pregnancy or BMI ≥ 35 kg/m^2^, an OGTT was also performed in GW 12. If random capillary P-glucose in GW 10, 25, 29, or 32 was >8.9–12.2 mmol/L, the woman was recommended to have an OGTT within the following seven days. If random capillary plasma glucose was ≥12.2 mmol/L, the woman was considered to have diabetes mellitus. Oral and written information were given, and if oral consent was obtained, inclusion took place. Women with gastric bypass surgery or inability to understand both spoken and written Swedish were excluded.

Data concerning the women’s age, prepregnancy BMI, weight, parity, GW at the time of OGTT, and indication for OGTT were manually extracted from the computerized patient record system (Obstetrix, Cerner Nordics, Sweden).

This study was approved by the Regional Ethical Review Board in Linköping, Sweden (2016/498-32). The date of approval was 29 November 2016. Consent from the study participants was given verbally and confirmed by taking a sample after the study information was given.

Data were divided into two parts with similar distribution concerning time for inclusion and with similar proportions of samples collected in different sampling orders. The first data set (*n* = 88) was used to calculate a conversion factor between venous and capillary P-glucose at fasting, and 1 h and 2 h after per oral glucose load, which was used to correct the capillary P-glucose cutoffs for the GDM diagnosis. The second data set (*n* = 87) was used to evaluate sensitivity, specificity, positive predictive value (PPV), negative predictive value (NPV), accuracy, and positive and negative likelihood ratios (LHR).

The OGTT was performed by five nurse assistants. The sampling was carried out in the following sequence: after an overnight fast, duplicate blood samples were collected both by capillary sampling from the sides of the third or fourth fingertips and by venous puncture of the ante-cubital vein. The women were numbered from 1 to 175. For women with odd numbers, capillary samples were taken before the venous samples, and for women with even numbers, samples were taken in the other order. The venous samples were collected in vacuum tubes spray coated with K2 EDTA (Vacutainer™, Becton-Dickinson, Franklin Lakes, NJ, USA) and were mixed 10 times by vertical inversion. Within 10 min, blood was absorbed on two test strips and immediately analyzed on the Accu-Chek Inform II (Roche Diagnostics Scandinavia AB, Solna, Sweden). A sample volume of 0.6 µL was required. Capillary samples were collected after wiping off two to three drops of blood from the side of the fingertip and measured using an Accu-Chek Inform II. Duplicate samples were taken. Subsequently, 75 g anhydrous glucose dissolved in 200 mL water was given for oral ingestion within five minutes. The sampling and measurement procedures were then repeated after 1 h and 2 h.

The Accu-Chek Inform II is a handheld device approved for diagnostic use that determines plasma glucose concentration by means of glucose test strips with a measuring range of 0.6–33.3 mmol/L. The glucose results of both the capillary and the venous determination are reported in mmol/L. The test principle of the Accu-Chek Inform II is based on direct measurement of glucose in the plasma phase using glucose dehydrogenase and pyrroloquinoline quinone (PQQ) as a coenzyme. After further reaction steps, this leads to formation of electrons which are measured amperometrically by the instrument. The analysis is not dependent on oxygen saturation in the sample and gives reliable results in the hematocrit interval of 0.10–0.65. The calibration of each test strip lot is traceable to the National Institute of Standard and Technology standards, and the reference method is based on isotopic dilution gas chromatography-mass spectrometry (IDGC-MS).

A total of four Accu-Chek Inform II instruments were used. Analytical quality was regularly checked by analysis of internal controls on two levels. The coefficients of variation (CVs) calculated for all these instruments during a 12-month period were 4.1% (level: 2.4 mmol/L, *n* = 178) and 2.8% (level: 16.0 mmol/L *n* = 171). New lots of test strips were checked on three mentor instruments which were also included in an external control program. In this program, results from individual laboratories are compared to values determined by a replicate of analyses on a reference method (IDGC-MS) or a reliable hospital laboratory method. The mean bias during the study period was −2.6% (*n* = 50) and all but one sample differed by less than 10% against the reference value. Upon introduction of AccuChek Inform II and prior to this study, validation was performed against IDGC-MS at Linköping University Hospital. Thirty plasma samples with different levels of glucose were analyzed on four different meters using one lot of test strips. Analyses were made in duplicates on each glucose meter and by the reference method. The method is based on derivatization of glucose to its aldonitrile pentaacetate, and ^13^C_6_-glucose was used as an internal standard. Briefly, analysis was performed on a gas chromatograph equipped with a CP-Sil-13 CB WCOT fused silica column (Chrompack International, BC, Middleburg, The Netherlands) and a HP MSD 5970 mass spectrometer (Hewlett-Packard, Palo Alto, CA, USA). Glucose was quantified using acquisition of ions at *m*/*z* 314.0 and 242.1 for unlabeled glucose and ions at *m*/*z* 319.0 and 246.1 for ^13^C-labeled glucose. For more details of the method, see Hannestad et al. [13].

### Statistical Analysis

To detect a difference in fasting plasma glucose at a minimum of 0.15 mmol/L with a correlation coefficent of 0.7, a standard deviation of 0.9, alpha <0.05, and a power of 0.8, sample size was determined to 172 individuals.

Normal distribution was checked using a Kolmogorov–Smirnov test. Normally distributed data are reported as mean ± SD. The significance of differences between capillary and venous samples was examined by a two-sided paired Student’s *t*-test. A *p*-value <0.05 was considered significant.

Capillary and venous P-glucose were further compared by a Passing–Bablock regression analysis. Bland–Altman plots were used to assess the analytical performance of the POCT device. This method was chosen because it is seen as the most appropriate way of determining limits of agreement between measurements [14].

Spearman´s rank correlation coefficient (*r*) was used for calculating correlation between all venous and capillary P-glucose values. The use of capillary sampling compared to venous sampling was assessed by calculating the sensitivity, specificity, positive and negative predictive values, accuracy, and likelihood ratios based on true positive, false negative, true negative, and false positive values. Venous P-glucose was looked upon as the standard (true positive) since recommended cutoffs are based on venous sampling [1].

The imprecision of P-glucose measured in capillary and venous samples was calculated from the values of duplicate samples using Dahlberg´s formula, and it is presented as relative standard deviation (CV%).

## 3. Results

The characteristics of the study population (*n* = 175) are shown in Table 1.

A total of 16 out of 2100 samples were missing (0.8%) due to technical sampling difficulties or interruption of the OGTT due to nausea.

Validation of Accu-Chek Inform II against IDGC-MS as a reference method showed a mean bias of −1.7 % and a high level of agreement for the whole measuring range reflected in a slope of 1.01 and an intercept of −0.18; see Appendix A. Only 2 out of 240 individual results on Accu-Chek Inform II differed by >10% from the reference method.

The imprecision calculated from the two measurements at each OGTT time point was higher in the capillary samples compared to the venous samples as seen in Table 2.

Capillary P-glucose measured by the Accu-Chek Inform II was significantly higher than venous P-glucose at fasting (4.99 ± 0.41 vs. 4.77 ± 0.41, *p* < 0.001), 1 h (8.38 ± 1.71 vs. 7.26 ± 1.70, *p* < 0.001), and 2 h (7.11 ± 1.61 vs. 6.21 ± 1.61, *p* < 0.001). The glucose values represent means from duplicate measurements and were normally distributed.

Passing–Bablock regression applied on venous and capillary P-glucose are shown in Figure 1a–c. The slope was not significantly different from 1 at any time point. A positive intercept was found for capillary samples at all time points in accordance with calculated mean differences. Capillary and venous P-glucose samples were significantly correlated with *r* of 0.93 (*p* < 0.001).

Data was divided into two sets as described in the Material and Methods section. Using the first data set, the difference between venous and capillary P-glucose at all-time points were analyzed by Bland–Altman plots and are shown in Figure 2a–c.

The mean differences were 0.22 mmol/L (95% CI: 0.18–0.27) at fasting, 1.12 mmol/L (95% CI: 0.99–1.26), at 1 h and 0.87 mmol/L (95% CI: 0.76–0.98) at 2 h after OGTT. Based on these findings, the cutoffs for a GDM diagnosis using capillary samples were corrected from 5.1 to 5.3 mmol/L, 10.0 to 11.1 mmol/L, and from 8.5 to 9.4 mmol/L for the fasting, 1 h, and 2 h samples, respectively.

The second data set was used to evaluate the corrected cutoffs obtained from the first data set by calculating the sensitivity, specificity, PPV, NPV, accuracy, and positive and negative LHRs, which are shown in Table 3.

Table 4 shows the number of women with a diagnosis of GDM in correlation to different sample methods and time points.

## 4. Discussion

We found a difference between capillary and venous samples of 0.2 mmol/L, 1.1 mmol/L, and 0.9 mmol/L in P-glucose during fasting, 1 h, and 2 h. Using capillary samples at fasting and after 2 h for the OGTT is an acceptable method for use in the antenatal care clinic considering the accuracy of 90.3% and the positive and negative LHRs of 16.4 and 0.2, respectively.

In accordance with our results, Ignell et al. [7] found a similar difference of 0.2 mmol/L (SD: 0.3) in fasting samples from 55 women previously having gestational diabetes. They found a greater difference between capillary and venous P-glucose 2 h after an OGTT, 1.5 mmol/L compared to 0.9 mmol/L in our study. However, their method was based on measurements in hemolyzed whole blood mathematically converted to P-glucose (HemoCue) in a nonpregnant population. Lee et al. [15] found a difference of 0.21 mmol/L (95% CI: 0.08–0.34) between venous and capillary fasting P-glucose samples measured simultaneously by two different glucose meters in 43 patients with diabetes mellitus type 1; although the population differs from ours, the result is similar to our study. Glucose levels vary depending on the source of the blood sample used for analysis; this variation is attributed to differences in glucose extraction by tissues, perfusion, oxygenation, pH, and temperature [16]. The increased volume of distribution in pregnancy may further affect these measurements [17], and it is possible that pregnant women have an altered glucose metabolism due to changes in hormones and metabolism [18]. In a study by Kruijshopp et al. including 2715 men and women undertaking an OGTT, they found a difference between capillary and venous P-glucose in fasting and at 2 h of 0.18 mmol/L and 1.09 mmol/L, respectively, and a sensitivity and specificity of 84% and 98%, respectively, for the diagnosis of type 2 diabetes mellitus using capillary sampling [19]. These are similar results compared to our study where we found a sensitivity and specificity of 85% and 95%. Furthermore, we found an 81% sensitivity, a 94% specificity, and an 85% PPV for using fasting P-glucose capillary sampling. These results are supported by a study of Coetzee where 122 women underwent OGTT eight weeks postpartum; they found a sensitivity of 89.3%, a specificity of 97%, and a PPV of 89% in fasting samples when comparing capillary and venous sampling. Similar agreements with our study were also found for the capillary sampling at 2 h [10]. The positive and negative LHRs of 16.4 and 0.2 indicate that capillary sampling at fasting and 2 h is reliable since a positive LHR above 10 indicates a high probability for correct diagnosis. LHRs are not dependent on disease prevalence in the group examined, which also indicates that capillary P-glucose measured by the Accu-Chek Inform II can be used safely for GDM diagnostics in other populations than ours.

The major strengths of this study were that duplicate venous and capillary samples were analyzed on the same instrument and that each instrument and lot of test strips were checked regularly in a quality assurance program that also included external controls. Accu-Chek Inform II was also validated against a reference method (IDGC-MS) prior to this study, which confirmed the high analytical accuracy previously reported. The large cohort of pregnant women allowed us to calculate the association between capillary and venous sampling, and we alternated between starting with the capillary and the venous samples in order to avoid making sampling order a bias.

The study has some limitations as only women that were able to understand both written and spoken Swedish were included, considering that women of non-European origin normally constitute a large group of women with GDM. On the other hand, there is no reason to suspect that this would compromise the results although the generalizability might be reduced [20].

A minority of women in this study underwent OGTT before gestational week 24. We used the same diagnostic criteria for GDM in this population. This might have affected the results, but as far as we know, there are no specified criteria for the diagnosis for GDM in the first trimester. On the other hand, it could be of importance to include all pregnant women regardless of gestational week since OGTT is performed both in early and late pregnancy.

We showed that capillary compared to venous samples had slightly greater imprecision as calculated from the results of duplicate sampling. This can probably be explained by the sampling technique, as P-glucose in capillary blood was measured in two different samples, whereas venous blood was taken from one and the same sample. As a general goal, analytical imprecision should not exceed half of the within subject variation (CV_I_ 5–6%) [21,22]. The goal for analytical imprecision was set at <2.7%. Single measurements resulted in an imprecision slightly above this goal. We therefore recommend duplicate samples when using Accu-Chek Inform II, which is in line with several other studies that have shown similar results as ours when duplicate samples were taken [7,8,23].

Our findings show that capillary sampling for measuring P-glucose in Accu-Chek Inform II POCT can be performed with high accuracy in an antenatal care setting using a risk-based screening with OGTT for the diagnosis of GDM. There are several advantages with capillary sampling, as it is easier to perform and requires simpler equipment. By using POCT, errors caused by glycolysis due to delayed measurement at a laboratory are reduced. The test result is also instantly available to facilitate further decision-making and the possibility to start treatment at once [10]. Daly et al. concluded that POC capillary maternal glucose tests were superior to customary laboratory practices for diagnosing GDM and that adjusted point-of-care glucose measurements have potential in the diagnosis of GDM [24]. O´Malley et al. compared POCT-corrected capillary glucose results versus laboratory venous plasma glucose results in OGTT for the diagnosis of GDM. They recommend the use of laboratory analysis when strict adherence to preanalytical sample handling can be applied; otherwise, POCT may be used for the diagnosis of GDM [25]. A recent Swedish study supported continuous use of POCT with the Hemocue device in GDM diagnostics [26]. Another recent study suggested that POCT using Accu-Chek can be used to obtain fasting values and to reduce overall waiting times for patients [27].

Additionally, the OGTT can be stopped when the criteria of GDM are reached according to the fasting glucose result, which in our study were one-fifth of the women. There are also rural areas globally where capillary sampling is the only option available and the International Federation of Gynecology and Obstetrics (FIGO) guidelines recommend use of a POCT glucose meter for the diagnosis of GDM [16]. One might speculate that the use of POCT also could be an attractive option during the ongoing pandemic of Covid-19.

Today in the clinical routine, most often only the fasting and 2 h glucose samples are analyzed when performing OGTT and there is no consensus regarding cutoff values and sampling types used for the diagnosis of GDM [28,29].

In this study, about one-quarter of the women were found to have GDM using the WHO 2013 guidelines [5]. The majority of the women performing an OGTT did so due to BMI ≥30 kg/m^2^ or a family history of diabetes type 2. In a risk-based screening strategy, the likelihood of having GDM among the women screened is relatively high and it is important not to miss any cases. The disadvantage of receiving a false-positive diagnose is minor since the treatment is based on recommendations concerning a healthy lifestyle and eating habits. The accuracy for using the fasting and 2 h samples with corrected cutoffs for capillary samples in our study is high, and the number of women receiving the diagnosis of GDM is almost identical in both sampling methods (40 versus 41 women).

## 5. Conclusions

We suggest that capillary fasting and 2 h P-glucose samples analyzed on the Accu-chek Inform II could be used for diagnosis of GDM during pregnancy using corrected cutoffs with acceptable accuracy in an antenatal care setting. Duplicate samples are necessary to maintain adequate precision. It is also worth continuing with OGTT when the fasting samples are within a normal range, since more women will receive a diagnosis of GDM.

## Figures and Tables

**Figure 1 diagnostics-10-01011-f001:**
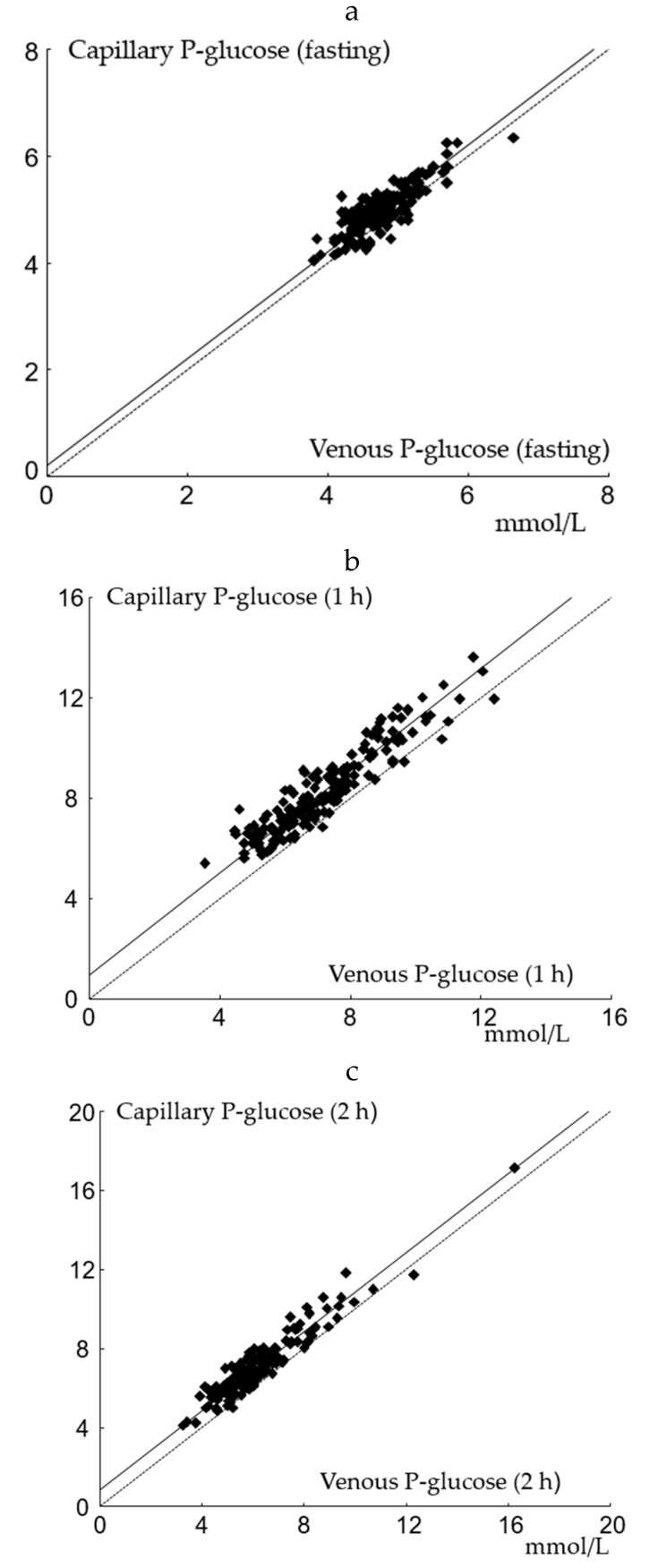
Passing–Bablock regression analysis of capillary and venous P-glucose. (**a**) fasting: the slope is at 1.00 (95% CI: 1.00 to 1.12), and the intercept is at 0.20 (95% CI: −0.37 to 0.20); *n* = 175; (**b**) 1 h: the slope is at 1.02 (95% CI: 0.96 to 1.09), and the intercept is 0.94 (95% CI: 0.45 to 1.37); *n* = 171; (**c**) 2 h: the slope is at 1.00 (95% CI: 0.94 to 1.07), and the intercept is 0.85 (95% CI: 0.42 to 1.17); *n* = 170.

**Figure 2 diagnostics-10-01011-f002:**
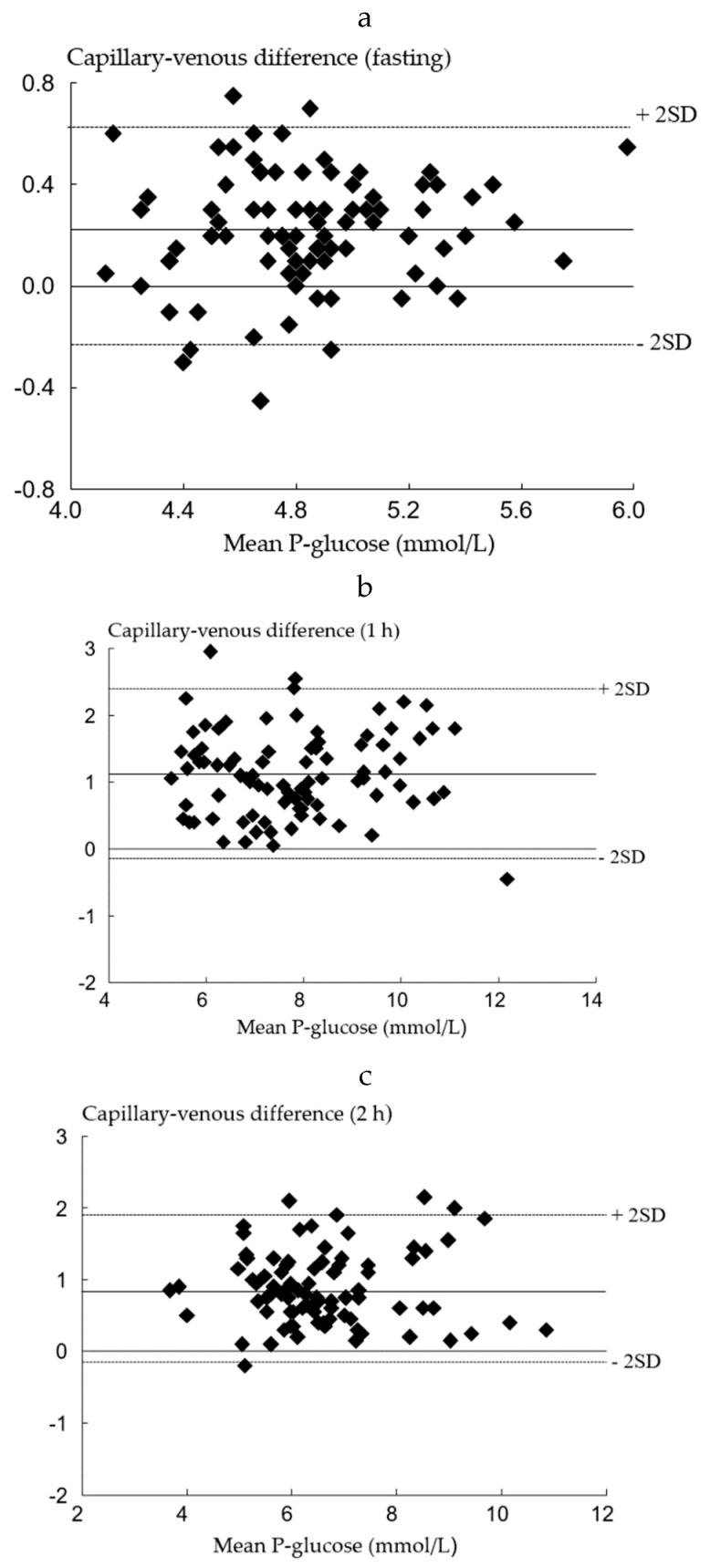
Bland–Altman plots of capillary and venous P-glucose in the first data. (**a**) set in fasting (*n* = 88); (**b**) set at 1 h (*n* = 87); (**c**) set at 2 h (*n* = 87).

**Table 1 diagnostics-10-01011-t001:** Characteristics of the study population (*n* = 175).

Characteristic		
Age (year) mean (SD)	-	31.5 (5.0)
OGTT in Gestational week	-	-
Min-max (median)	-	11–34 (28)
*n* (%)	<24 weeks	26 (14.9)
	≥24 weeks	149 (85.1)
BMI (kg/m^2^) *n* (%)	<20.0	5 (2.9)
	20.0–24.9	40 (22.9)
	25.0–29.9	40 (22.9)
	30.0–34.9	49 (28.0)
	35.0–39.9	33 (18.9)
	≥40.0	8 (4.6)
Parity *n* (%)	nulliparous	63 (36.0)
Indication for OGTT * *n* (%)	BMI	90 (51.4)
	Family history of type 2 diabetes	56 (32.0)
	Previous LGA	13 (7.4)
	Previous GDM	7 (4.0)
	Non-European origin	3 (1.7)
	≥8.9 mmol/L, random glucose	5 (2.9)
	Previous IUFD	1 (5.7)
	Other	5 (2.9)

LGA = Large for gestational age; GDM = Gestational Diabetes Mellitus; IUFD = Intrauterine fetal death; BMI = Body Mass Index; OGTT = Oral Glucose Tolerance test. Other includes women with oocyte donation, polyhydramnion, and unspecified. * One woman could have more than one indication for OGTT.

**Table 2 diagnostics-10-01011-t002:** Evaluation of imprecision based on capillary and venous duplicate samples.

Sampling Point	Coefficient of Variation (%) Venous	Coefficient of Variation (%) Capillary
Fasting, single value (*n* = 175)	2.48	3.37
OGTT 1 h, single value (*n* = 171)	2.79	3.66
OGTT 2 h, single value (*n* = 170)	2.51	3.43
Fasting, mean from duplicate (*n* = 175)	1.75	2.38
OGTT 1 h, mean from duplicate (*n* = 171)	1.98	2.59
OGTT 2 h, mean from duplicate (*n* = 170)	1.77	2.43

**Table 3 diagnostics-10-01011-t003:** Sensitivity, specificity, positive predictive value (PPV), negative predictive value (NPV), accuracy, and positive and negative likelihood ratios (LHR) for corrected cutoffs in capillary sampling with venous sampling looked upon as the standard (true positive).

Test Corrected Cut-Offs for Capillary Samples	Sensitivity (%)	Specificity (%)	PPV (%)	NPV (%)	Accuracy (%)	LHR (Positive)	LHR (Negative)
Fasting ^a^	81.0	95.4	85.0	94.0	92.0	17.8	0.2
1 h ^b^	71.4	97.4	71.4	97.4	95.2	27.5	0.3
2 h ^c^	100	98.7	85.7	100	98.8	77	0.0
Fasting ^a^, 1 h ^b^ and 2 h ^c^	88.1	92.5	78.7	96.1	91.4	16.4	0.2
Fasting ^a^ and 2 h ^c^	85.0	95.0	83.0	96.0	90.3	16.4	0.2

ᵃ ≥5.3 mmol/l; ᵇ: ≥11.1 mmol/l; ᶜ ≥9.4 mmol/l.

**Table 4 diagnostics-10-01011-t004:** Number of women diagnosed with gestational diabetes mellitus in relation to sampling method and OGTT time points (*n* = 175).

Sampling Method	Fasting Sample *n* (%)	One-Hour Sample *n* (%)	Two-Hour Sample *n* (%)
Venous samples	36	11	11
Capillary samples *	35	15	14
Cumulative number venous samples	36 (20.6)	40 (22.9)	44 (25.1)
Cumulative number capillary samples *	35 (20.0)	44 (25.1)	47 (26.9)
Cumulative number venous samples ^#^	36 (20.6)	X	40 (22.9)
Cumulative number capillary samples *^,#^	35 (20.0)	X	41 (23.4)

* Corrected cutoffs were used for capillary samples # Not taking the 1 h sample in to account.

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
