# Peer review of "Comparison of Venous and Capillary Sampling in Oral Glucose Testing for the Diagnosis of Gestational Diabetes Mellitus: A Diagnostic Accuracy Cross-Sectional Study Using Accu-Chek Inform II"

_diagnostics, 2020, doi:10.3390/diagnostics10121011_

Round 1

Reviewer 1 Report

Dear Authors,

I have read with great interest your article.

Diagnosis of gestational diabetes is stil controversial. Authors suggest that capillary fasting and 2 h P-glucose samples analyzed on the Accu-chek Inform 295 II could be used for the diagnosis of GDM during pregnancy using corrected cut-offs with acceptable 296 accuracy in an antenatal care setting. 

The objective of the study is clearly stated. Manuscript is written clearly. Study limitations are listed clearly as well. I have no objections. 

Author Response

Reviewer 1

Dear Authors,

I have read with great interest your article.

Diagnosis of gestational diabetes is stil controversial. Authors suggest that capillary fasting and 2 h P-glucose samples analyzed on the Accu-chek Inform 295 II could be used for the diagnosis of GDM during pregnancy using corrected cut-offs with acceptable 296 accuracy in an antenatal care setting. 

The objective of the study is clearly stated. Manuscript is written clearly. Study limitations are listed clearly as well. I have no objections.

Thank you very much for your review and comments.

Reviewer 2 Report

The manuscript entitled "Comparison of Venous and Capillary Sampling in Oral Glucose Testing for the Diagnosis of Gestational Diabetes Mellitus: A Diagnostic Accuracy Cross-sectional Study using Accu-Chek Inform II" aimed to study if  plasma glucose cut offs for Gestational Diabetes Mellitus diagnosis based on venous sampling can be replaced by cut offs based on capillary sampling without compromising diagnostic accuracy.

Overall, the manuscript is well written, with a solid introduction, a very complete materials and methods section and clearly presented results. 

Regarding the abstract section, it could be more complete, specifically regarding the background of the work. 

Attention should be paid to the fact that the term Gestational Diabetes Mellitus appears first in the abstract as GMD however it is not explained that GMD is an abbreviation to it. The same happens in the introduction. 

The main concern regarding the manuscript is the novelty of the theme since there are several studies comparing the possibility to use capillary vs venous sampling in OGTTs, for the diagnosis of GMD. Some of them were mentioned in the discussion of the manuscript, but there are a few that were not discussed. 

Author Response

Reviewer 2

The manuscript entitled "Comparison of Venous and Capillary Sampling in Oral Glucose Testing for the Diagnosis of Gestational Diabetes Mellitus: A Diagnostic Accuracy Cross-sectional Study using Accu-Chek Inform II" aimed to study if plasma glucose cut offs for Gestational Diabetes Mellitus diagnosis based on venous sampling can be replaced by cut offs based on capillary sampling without compromising diagnostic accuracy.

Overall, the manuscript is well written, with a solid introduction, a very complete materials and methods section and clearly presented results. 

Regarding the abstract section, it could be more complete, specifically regarding the background of the work. 

Response 1: We have added a sentence about the background and have made minor adjustments. The abstract is now 202 words, we hope this is to your satisfaction.

Attention should be paid to the fact that the term Gestational Diabetes Mellitus appears first in the abstract as GMD however it is not explained that GMD is an abbreviation to it. The same happens in the introduction. 

Response 2: Thank you for the comment, we have adjusted this in the manuscript.

The main concern regarding the manuscript is the novelty of the theme since there are several studies comparing the possibility to use capillary vs venous sampling in OGTTs, for the diagnosis of GMD. Some of them were mentioned in the discussion of the manuscript, but there are a few that were not discussed.

Response 3:Thank you very much for your comment. We have found an additional reference from 2020 which we hade added to the manuscript. O´Malley et al have compared capillary glucose in point-of-care-testing to a labaratory venous plasma glucose testing during OGTT in pregnant women. The novelty with our study is that we have compared capillary and venous sampling both analyzed on Accu-Chek Inform II (point-of-care-testing). To our knowledge, this has not been published before.

Accu-Chek Inform II was validated against a reference method previous to our study and is only used by medical staff. The main advantage of using Accu-Chek Inform II this way is that errors caused by glycolysis due to delayed measurement at a laboratory are reduced.